# Dancing with the Sniper: Rasha Abbas and the "Art of Survival" as an Aesthetic Strategy

Moritz Schramm 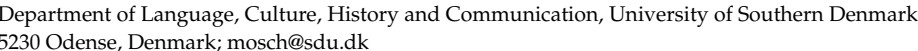

Department of Language, Culture, History and Communication, University of Southern Denmark, 5230 Odense, Denmark; mosch@sdu.dk

**Abstract:** In the last few decades, a growing dissatisfaction with traditional approaches can be observed in migration and refugee studies. In particular, the widespread focus on the "refugee" and "migrant" as exclusive objects of study has been criticized for its underlying tendency of repeating the binary polarization between migrant and non-migrant, native and foreign as well as majority and minority. This chapter considers the short stories of Syrian journalist and writer Rasha Abbas against this background. Instead of reducing her stories to the depiction of flight and exile, this chapter explores her stories as aesthetic expressions of what can be called the "art of survival"—the concept focusing on strategies of empowerment and tactics to regain autonomy. In Abbas' prose, this "art of survival" is achieved and expressed through the blending of times and spaces as well as the aesthetic transformation of reality into surreal realms. Experiences of war, displacement, exile, and patterns of exclusion in the new homeland merge into complex pictures of the human capacity to reframe and reinvent a given reality. When viewed from this perspective, the surreal and psychedelic nature of her writing intensifies the power of aesthetic freedom, thus helping overcome traditional representations of migrants and refugees in cultural expressions and literature.

**Keywords:** art of survival; Rasha Abbas; aesthetic freedom; postmigration; post-other; victimization; maxim gorki theater; reflexive turn

## 1. The "Reflexive Turn" in Migration Studies

In an influential essay from 2016, migration scholar Janine Dahinden observed how a growing number of voices in the migration studies community had been calling for "more reflexivity on the part of migration researchers" (Dahinden 2016, p. 2208). This call is sometimes also described as a "reflexive turn" and is a reaction to what Boris Nieswand and Drotbohm (2014, p. 1) referred to as an "intellectual crisis" in migration studies. Moreover, since the late 1980s, categories such as "ethnicity", "society," and "culture" have lost their conceptual innocence, and even that of "migrant" has increasingly been rejected as an assumingly neutral description.[1] According to Dahinden (2016, p. 2209), this category is fundamentally bound to the logic of the nation state and the "supposedly natural congruence" among different national, territorial, political, cultural, and social boundaries. Studies on migration and exile grew out of this historical foundation and are arguably still bound to the same "paradigm of normalized difference" (Dahinden 2016, p. 2210). Consequently, Dahinden (2016, p. 2210) noted the "difference" between migration and non-migration as "ultimately the *raison d'être* of migration research". On a methodological level, this intellectual crisis has incited ongoing discussions on how to reframe migration studies when seeking to avoid the danger of unwillingly reaffirming traditional boundaries and distinctions through the scholarly focus on "migrant" and "refugee" as the only objects of scholarly interest. In this context, the cultural anthropologist Regina Römhild (2017, p. 70) even identified a "fundamental dilemma" for critical migration research. While on the one hand, this research seeks to classify migration as "a productive societal and cultural force" to counter anti-immigration discourses in the public sphere, on the other, it runs

the risk of reproducing the very distinctions it wants to overcome (Römhild 2017, p. 70). In particular, the widespread strategy of endlessly repeating the "narrative of alternative, transnational, hybrid migrant worlds" runs the risk of again fixing the "migrant" outside the normality of society (Römhild 2017, p. 70). Thus, as Römhild argues in her essay from 2017, the underlying problem is:

> that migration research is often understood merely as "research about migrants", producing a "migrantology" that is capable of little more than repeatedly illustrating and reproducing itself; a "migrantology" that at the same time plays its part in constructing its supposed counterpart, the national society of immobile, white non-migrants. (Römhild 2017, p. 70)

However, Römhild's answer to this dilemma has gained traction in part of migration research. Along with Manuela Bojadžijev and others, she advocated for a methodological turn that she described as the need to demigrantize migration research while migrantizing it into culture and society (Römhild 2017, p. 70; see also: Bojadžijev and Römhild 2014; Dahinden 2016; Petersen et al. 2019, pp. 13–15; Yildiz 2022). Instead of making migration itself the object of study, she advocated for a research perspective that "takes as its starting point in societies negotiations over migration" (Römhild 2017, p. 72). Further, she explained that what is lacking "is not yet more research about migration, but a migration-based perspective to generate new insights into the contested arenas of 'society' and 'culture'" (Römhild 2017, p. 70). Dahinden's (2016) appeal for the "demigrantization" of research on migration and integration is in a similar direction.[2]

## 2. The Politics of Representation

In literary and cultural studies, the aforementioned appeal for the "demigrantization" of migration studies has been echoed rather slowly. Instead, for a long time, there has been a strong focus on literature written by migrantized persons and their descendants, often labeled as "migration literature" or the literature of migration. Historically, this influential label grew out of the attempt to disrupt persistent narratives of national homogeneity and, in opposing them, recognize the growing cultural, ethnic and religious diversity of contemporary societies (Schramm 2018). And even if the "migration literature" label is less used in more recent approaches, some elements of its critical approach are still visible. In particular, major parts of the current studies on the representation of migration and flight are focusing on the "voice of the refugee" and the "perspective of the refugee" (Bromley 2021, p. 59). This approach is typically employed as a critical intervention against the longstanding tradition to represent "migrants" and "refugees" as either threats or victims in public discourse. Since the so called "refugee-crisis" in 2015, the public discourse in most of Europe has been dominated by a securitization-approach, where forced migration is mainly viewed in relation to questions concerning security and border control. In this context, the perspectives of persons with a background in migration and flight has been "largely ignored" (Hill 2019, p. 302). The most recent focus on the role of art and literature must be considered against this background. Specifically, as cultural studies scholar Roger Bromley (2021, p. 9) pointed out, art and literature are often seen as mediums that can help provide agency to the "refugee" or "migrant" by creating space for the "perspectives of the refugee," exploring "counter-narratives", and subverting or at least questioning hegemonic and nationalistic narratives as well as "negative, populist representations of refugees" (See also: Hallensleben 2021, p. 197). There is no doubt that this focus on the "perspective of the refugee" is necessary and extremely important, not in the least to give more nuance to the public discourse. However, in academia, it is often overlooked that this "epistemological turn" (Yildiz 2022, p. 45) cannot necessarily and by itself overcome the "fundamental dilemma" of critical migration studies, referred to above. Rather, the focus on the aforementioned perspective also implies, by positioning the "refugee" as the Other, the risk of reproducing the binary distinctions between the migrant and non-migratory we-group. This becomes especially apparent when the focus on the voice of the refugee is combined with the expectation of biographical coherency, truthfulness, and authenticity.

In this context, the theater critic Nora Haakh (2013, p. 38) identified a widespread tendency of writers and artists with "immigration backgrounds" being expected to deliver personal and authentic insight into supposedly "shuttered worlds" to non-migratory readers. This "cult of authenticity" (Haakh 2013, p. 38) not only limits the range of stories that the "refugee" and "migrant" is supposed to tell but also reaffirms and reproduces the persistent distinction between "us" and "them" and between the supposedly non-migratory we-group and the Other. Conclusively, the focus on the "perspective of the refugee" is not in itself enough to overcome persistent patterns of representation and anti-immigration narratives. Rather, this important approach has to be reflected upon and discussed in relation to the aesthetic dimension of the story as well as the subject positions the "refugee" is expected to take. The issue here is how to create space for marginalized voices and perspectives while simultaneously avoiding reproducing traditional representations and ascribing the "migrant" and "refugee" labels to predefined positions in society.

In an essay from 2022, the Syrian actress Kenda Hmeidan discusses a similar issue by focusing on the challenges that former refugees from Syria are confronted with in their new homelands. Everybody she knows who like herself fled from Syria to Europe in recent years had a different story about how they ended up in the countries where they are currently living in exile, and everybody has "his own tools how to settle, how to start to integrate or refuse to integrate within society" (Hmeidan 2022, p. 4). However, they are all involved in the same transition from "being a refugee to being a citizen" (Hmeidan 2022, p. 4). In this process, Hmeidan emphasized that dealing with the memory of the past is particular challenging. While the focus on the memories of the past arguably allows the exiled Syrians to "save" the lost homeland, it also implies the risk of remaining fixed on the past and thus preventing oneself from moving forward and redefining one's own identity. In this context, Hmeidan (2022, p. 4) asks, "Will forgetting helps us to move on? Or is it the other way around? [ . . . ] When will we stop digging inside us and will have the opportunity to look around us?".

Here, it is clear that these questions do not just simply refer to a traditional narrative of "arrivedness", which would include the idea of a linear progression from the position of being a "migrant" or "refugee" towards the new position as an allegedly integrated or assimilated subject (Heidenreich 2015, pp. 300–1; see also Petersen 2019, p. 79). For Hmeidan, the challenge of how to deal with the past instead implies the formal and aesthetic challenge of when and how to tell the story of the past as well as of how to include this story in a future identity without being fixed in a specific position in the society. Hmeidan (2022, p. 5) goes on to ask, "In what forms can Syrians express their stories, ideas, trauma, and their transitions from the homeland to Europe?" before adding:

> How can we talk about our experiences, without always being in the position of the victim, "the poor Syrian refugee"? How can we talk about the past, not by showing only the drama and the destruction where we came from (and the constant nostalgia of leaving it) but about a past which can enrich the present, and the future? (Hmeidan 2022, p. 5)

Here, Hmeidan's questions are not only about the memorization of the past but also about possible subject positions and identities in society. The focus on the past seems to limit and restrict former refugees from moving forward and from, in particular, overcoming the persistent victimization of "refugees" and "migrants" in public discourse, which, once again, contributes to the distinction between the "refugee" on the one side and the supposedly white and non-migratory we-group on the other. Moreover, Bromley (2021) addresses the challenges connected to this victimization. In his attempt to develop counter-narratives to traditional, nationalistic narratives about refugees and migrants, he also turns to the narrative of the refugee as the "sentimentalized, passive victim, the object of compassion" (Bromley 2021, p. 8). The widespread humanitarian concern and sympathy in public discourse tends to focus on the vulnerability of the refugee. And while this vulnerability should obviously be protected, the tendency to see all refugees as victims or vulnerable people "is a perspective that needs to be critically examined for

its reductiveness and refusal of agency" (Bromley 2021, p. 8). The challenge here is how to find new complex representations, which allow for "the agential subject, the resistant activist," and "the newly emergent citizen" (Bromley 2021, p. 8). In this context, one can even argue that every representation of flight, migration, and exile always is incoherent, as it always implies a level of distance from and misrepresentation of the sometimes-traumatic experiences of the past. This challenge does not necessarily disappear when the focus is on the marginalized voice of the Other or the perspectives of those who, according to the predominant Eurocentric worldview, have been and still are conceived of as "non-modern," "non-civilized," "non-enlightened," and "non-emancipated" (Yildiz 2022, p. 45).

As Butler (2005) pointed out, every attempt to give an account of oneself is necessarily bound to schemes and narratives, which are not possessed by the subject alone but rather exist outside of the subject. In relation to migration studies, this further increases the need for more multifaceted narratives and more complex representations, which are not bound to traditional expectations and hegemonic narratives as well as avoid falling back into "terminological ghettos" (Geiser 2015, p. 307). Bromley addresses these issues when he refers to the challenges of representation and the need to broaden our understanding of the possibilities and limits of what has been called the politics of representation (Hall 1997). While the "experience of refugees *is* unrepresentable in a sense" and "representational forms are always inadequate," Bromley (2021, p. 8) argues that new narratives and lenses are still necessary to create space for new perspectives and forms of representation. To do so, he mentions the "development of other lenses for perception," "a greater aesthetic-political reflexivity and sensitivity" as well as "a search for new, and radical, rhetorical forms which unsettle and disrupt expectations and preconceptions" (Bromley 2021, p. 8). In the end as Yildiz (2022, p. 45) noted, an "epistemological turn" is required, substituting the focus on traditional politics of representation with that on processes of "ambiguities and liminal experiences".

One specific way of dealing with the aforementioned challenges and offering new and more complex representations and aesthetics can be found, as I will argue in this chapter, in the literary works of the Syrian journalist and writer Rasha Abbas. Abbas was born in 1984 in Latakia, Syria, and left Syria in 2011 to Lebanon and three years later to Germany, where she currently lives in exile. In her short stories, she not only explores the consequences and aftermaths of war and forced migration but also develops an aesthetic strategy, which allows her to combine the past with the present and explore what I call the "art of survival". In this strategy, the subject regains autonomy and agency by rejecting persistent external ascriptions and replacing them with novel and imaginative escape routes, transforming the reality into a new space of negotiation and participation. In particular, the widespread public demand for authenticity and realism is turned upside down in Abbas' stories. Through her combining of times and places and focus on the inner multiplicity of persons and protagonists in her stories, she manages to redefine her own subject position beyond external expectations and ascribed labels. Further, her aesthetic strategy of the "art of survival" allows her to depict the cruelties and atrocities of war and exile on the one hand, while simultaneously making space for a broader space of negotiation and dealing with the challenges of memories and times on the other. This broader space of negotiation is also mirrored in relation to language. Abbas' works are not only part of a long-standing Syrian and Arabic literary tradition but have also started to circulate and be read in translation within Europe. Some of her books have even been published first in German before being published in Arabic, becoming in this way part of an ongoing negotiation between memory, space, and time within Europe. My own lack of Arabic language skills means that I have only read English and German translations of her texts. It mirrors, however, the transformations and openings, embodied by her work: instead of remaining fixed on the past, her work rather conflates times and spaces. The transition from being a refugee to "being a citizen", referred to above, involves moving between old and new languages and growing into a field of cosmopolitan literature that extends beyond binary distinctions. Before focusing on the theoretical and methodological

challenges of her aesthetic strategies, it is necessary to take a brief look into some elements of Abbas' writing and at the staging of her works at the Maxim Gorki Theater in Berlin at the beginning of 2022, which was also the point of departure for Hmeidan's essay.

### 3. War, Displacement, and Aesthetic Freedom

Rasha Abbas' first literary publication goes back to her time in Syria, where she lived as writer and journalist. In 2008, she won a young writers award in Syria for her first collection of short stories, *Adam hates TV*. However, in the wake of the Syrian revolution, she was forced to leave the country. Her breakthrough to a broader audience came in 2016 when she published her second book of short stories, *The Invention of German Grammar* (Abbas 2016). The book, originally written in Arabic, was published in the German translation before the original and was received positively by German critics and readers, not in the least because of her humoristic and critical stance on the challenges of arriving in a new country. Subsequently, she published several stories partly translated into English and German in magazines and newspapers.

In 2018, her next collection of short stories came out, once again, first in German: *Eine Zusammenfassung von allem was war*, which is *A Summary of What Happened* in English. The book has been a major success among critics as was its staging at the Maxim Gorki Theater in Berlin in the beginning of 2022. Unlike those from traditional mainstream literature, the stories are formally and thematically challenging. They often combine surreal and dreamlike scenarios with the description of war and its aftermath, thus allowing for new and different perspectives. In a short reader's report, the translator Guthrie (2017) characterized Abbas' literary stories as being "eclectic, intense, often psychedelic." Many of her stories, Guthrie (2017) continues, "are dreamscapes which creep up on the reader with sudden plunges into haunting hyper-realism, operating within a punk aesthetic". These dreamscapes include experiences of the Syrian war, dancing scenes, surrealistic parties, and family negotiations in unspecified places. They are obviously taking place in both her former homeland and in exile. Political events from the Syrian revolution and the war are intermingled with experiences of fleeing to a new homeland and living in exile. In various stories, we are confronted with the cruelty and harshness of war as well as with death, destruction, and the experience of being powerless when standing at checkpoints in warzones as well as at European borders. In other stories, these experiences are combined with surreal scenes such as the following: an overflowing toilet, which is flooding the narrator's house during her mother's family party due to a positive pregnancy test that she tried to flush down the toilet; or the depiction of a severed head, exhibited in a flowerpot in a private house, which the narrator is expected to water, so it can grow roots. The flap-text of the German edition attempts to provide an overview of some of the motifs: "Time-loops and Russian rockets above outdoor swimming pools, paranoid teenagers, checkpoints and remote hotels. Drug trips, cinema-productions for dictators and refugees living in exile in collective housings: in *A Summary of What Happened* the images of the old and the new home are merging into each other" (Abbas 2018, cover).

The stories are often situated in "dreamlike, gaming realities" but also take place in the "dusty-grey rural world of the Middle East, in tropical swamp landscapes, in endless deserts of ice, or in the villa of Max Liebermann," a famous German-Jewish painter who lived in Berlin in the beginning of the 20th century (Abbas 2018, cover).[3]

Abbas' surrealistic, psychedelic style was already visible in one of her early stories, *A Plate of Salmon is Not Completely Cleaned of Blood*. This story—translated from Arabic into English by Alice Guthrie and published in the broadly discussed volume *Syria Speaks: Art and Culture from the Frontline* in 2014—also combines different layers of reality and offers a level of aesthetic transformation of reality (Abbas 2014). The story starts out with the narrator's dialog with another person—seemingly directly addressing the reader—who is invited to eat salmon for lunch at the narrator's house. Even though the place where the story takes place is never mentioned directly, references to the Syrian war are evident. For example, the story directly refers to historical facts such as the massacres from Baniyas

and Al-Bayda, which received international attention after Syrian activists posted videos and pictures on social medias (BBC 2013). However, in the story, these historical events are portrayed with a focus on their appearance in social media. The narrator says "Baniyas? Of course, I am friends with it on Facebook," at one point in the story before referring to the videos documenting the atrocities committed by the Syrian regime (Abbas 2014, p. 276). As is typical in Abbas' writing, such references to warzone massacres merges with daily routines, often combining the horror of war crimes with daily-life experiences. Specifically, war experiences are interspersed with questions about the narrator's hobbies and smoking habits. "You don't smoke? My hobby is collecting foreign coins," the narrator says before continuing, "please, eat up, there is salmon for lunch—don't watch the corpses and children with ripped out fingernails on YouTube before lunch, you don't want to lose your appetite for salmon" (Abbas 2014, p. 276).

This merging of daily normality with killings and torture becomes particularly significant in relation to a sniper who is running through the house to kill the narrator's mother. In the beginning, the narrator asks the invited lunch guest to not tell the *Hajja* (an Arabic term referring to an older and respected woman, which later appears to be the narrator's mother) that she is smoking in the house: "Whatever you do, please don't tell the Hajja that I smoke in the house and that I invite my friend the sniper in while she is not there [ . . . ]. The sniper's taking a stroll along the corridor, come and listen to this Robbie William song with me" (Abbas 2014, p. 276).

Inviting the sniper into the house may be read as a literary symbol for the aftermath of the Syrian revolution. While the revolution was overwhelmingly begun by a young generation of Syrian activists, the regime then turned against them, thus threatening every single household and family member. Consequently, private spaces were transformed into war zones causing the narrator to blend the threats of war with not only daily routines but also private dreams and fantasies. The depiction of the sniper running through the house is related to the longing for a popular crime series on television and sexual fantasies ("it'd turn me on if you made love to me with the TV on", Abbas 2014, p. 278). Later, eating lunch while watching TV appears as a possibility in the narrator's reality, even after her mother had been killed by the sniper: "Listen, if the sniper kills my mum we'll go and eat that salmon with slices of lemon on her bed, there's a TV in there too, we'll watch it while we stain her clean sheets with a bit of salmon, what do you say to that?" (Abbas 2014, p. 278).

These references to the sniper, who potentially may kill the narrator's mother, seems to refer back to the well-known novel *The Story of Zahra* by the Lebanese author Hanan al-Shaykh, which was published in 1980. In this novel, the plot of which takes place during the Lebanese civil war, the narrator, Zahra, invites a sniper into her house even though she is afraid of being killed by him. While al-Shaykh's novel ends with the killing of the female narrator, Abbas' reworking of the story at least provides some agency back to the narrator. In contrast to the abuse and killing of Zahra in al-Shaykh's novel, the morbid and sarcastic option of eating lunch on the bed of the murdered mother offers a dreamlike vision of survival. Accordingly, the scene not only illustrates the shifting realities in Abbas' writing— combining TV-series, gaming realities, and sexual fantasies with war-experiences and existential threats—but also includes attempts at regaining control through the dreamlike transformation of reality, as shown by the following comment in the story. Confronted with the threat of the sniper, the narrator considers the possibility of stopping him by embracing him, slowing him down for a dance: "The sniper is running through the corridor, what a shame he doesn't slow down so we could dance with him a bit" (Abbas 2014, p. 278). Even though the dance never actually takes place in the story and even though it would require an active change in the sniper's behavior—he would have to slow down—the somehow strange picture of a common social activity with the potential perpetrator disrupts the traditional representation of war and its consequences. The picture also seems to refer back to a scene in al-Shaykh's ([1980] 1995, p. 157) novel, where Zahra considers whether a "troupe of dancers" would help to divert the sniper from aiming his rifle and randomly killing people. In both stories, the idea of dancing offers a disruption to the power of

war and violence. In Abbas' story, however, the picture also becomes surreal and even humoristic. The idea of "dancing together" minimizes the real threat by including it in the imagination of a common activity, in which the narrator and potential perpetrator meet and engage with each other on an assumingly equal level.

In her essay *Dreaming the Same Dream in Different Places*, which was previously quoted, the actress Hmeidan (2022) described the aforementioned scenes of aesthetic imagination as attempts in Abbas' prose to open new escape routes, thus giving agency back to the characters as well as to the reader. When she read Abbas' prose for the first time, Hmeidan noted that they were completely different from all other representations of war, flight, and exile that she has ever come across. Additionally, the strength of Abbas' writing, Hmeidan explained, can be found in her combining of different perspectives, times, and realities. Instead of being defined by the past, Abbas manages to transform topics such as exile, detention, isolation, loneliness, alienation, torture, and loss of identity into a new reality. The fact that Abbas chooses not to "document" the devastation of war "or talk about it from a personal perspective," makes it possible for her to depict the atrocities of war and challenges of the exile while simultaneously overcoming the danger of being defined by the memories of the past (Hmeidan 2022, p. 6). It is "through fictional and abstract images, by inventing surreal events and scenarios, using sarcasm and contempt" and with the "overlapping of places and times" that Abbas' stories suggest "the possibility of escaping" (Hmeidan 2022, p. 6). Hmeidan (2022) goes on to elaborate, saying:

> The constant movement of the characters from one place to another and the inability to stand still in a world, where chaos rules. By refusing to take reality as it is, it gives us as readers a feeling of resistance and the capability to survive what is happening, either through physical movement or mental moving our strong ability to imagine and to jump in our heads and times and places. (p. 6)

Both the protagonists of the story and the reader are emboldened to explore shuttered doors, new escape routes, and hidden paths through its realities. As in computer games where protagonists can choose different paths and versions of reality, Abbas' aesthetic approach gives the characters in the stories "an ability to choose" (Hmeidan 2022, p. 6). As it is the case in the early writings of German philosopher Walter Benjamin or in Gaston Bachelard's ([1958] 2014) reading of the transformative power of architecture and spaces in *The Poetics of Space*, the logic of the dream extends the idea of a clearly shaped and inescapable reality and thus creates space for the very possibility of social change and transformation. Following the writings of German philosopher Rebentisch (2016), one can even see similarities with the transformative and political power of the notion of "aesthetic freedom." More specifically, Rebentisch considers this notion as the foundation of all democratic existence. She goes on to argue that the experience of "aesthetic freedom" allows the individual to mediate between two different but yet interrelated dimensions of an individual's identity: dependency on pregiven social norms and values, which at least partly define personal identity; and the idea of complete freedom from the social order, which is often assumed to give agency to an individual. The tension between the two polarities—the uncritical acceptance of pregiven and ascribed identities on the one hand and the complete distance from all predefined social roles on the other—is constantly negotiated inside the individual. Typically, this tension appears as the "immediate experience of self-difference" (Rebentisch 2016, p. 9), which is that when the pregiven roles and identities do not match the experience of who we are or want to be. Abbas' artistic engagement with the war and exile and her usage of imaginary, dreamlike escape routes and alternative realities includes a similar confrontation between the pregiven social order and the possibility to define one's own life. The aesthetic distance from the social world is a necessary condition "for the self-determined appropriation or transformation of the social practices by which we are always already determined" (Rebentisch 2016, pp. 9–10). Abbas' dreamlike prose thus constantly redefines the political community as a concept.[4]

## 4. Post-Otherness and The Imaginative Formation of the Theater

At first glance, Abbas' surreal aesthetic and her insistence on "aesthetic freedom" may be seen as world escapism, where the characters are fleeing from political struggles and conflicts into a dreamlike world of fictive existence. Alternatively, as Halasa (2015, pp. 165–66) puts it, Abbas "uses fiction to affirm the value of the individual amid the collective barbarities of the conflict". Conversely, from a theoretical perspective, the collapse of stable representations and the combination of spaces and time can also be considered part of what curator Bonaventure Soh Bejeng Ndikung and cultural anthropologist Römhild famously called the emergence of "post-Otherness" and the figure of the "post-Other." (Ndikung and Römhild 2013). Further, they critically engage the historical and ongoing production of "Others" at various racist, sexist, and political levels (Ndikung and Römhild 2013, p. 207). In particular, the widespread notion of "integration,"—discussed here as the ideology of "integrationism"—is strongly connected to processes of Othering (Ndikung and Römhild 2013, pp. 211–12). It also plays a major role in establishing the idea of a supposedly homogenous, white we-group as the foundation of the modern nation state. The promise and the condition of integration, Ndikung and Römhild (2013, p. 213) argue, "makes use of the assimilated Other to create and stabilize the notion of a 'natural', racially unmarked, white self". This process of constructing and stabilizing an allegedly stable we-group is historically intertwined with the colonial thinking of spatial and temporal integration. However, this historical condition is changing due to more recent transnational openings and entanglements. While the "colonial Other" has historically been "integrated into the binary hierarchical relation between 'metropolis' and imperial 'periphery' across geopolitical distance", this "spatial order of 'here' and 'there'," according to Ndikung and Römhild (2013, pp. 213–14), "is collapsing because of the past and present of migrations and mobilities". In this situation, a paradoxical moment in history is clear. On the one hand, new forms of mobility and migration have led to a spatial implosion, where the formerly distant Other is now flourishing inside Europe. Former and ongoing migration movements have not only changed the components of populations in European countries but also incited new forms of self-perception, new identity formations, and new ways of defining one's belongings (Bromley 2006; Römhild 2018). Consequently, Europe now consists of "a multitude of minorities" (Ndikung and Römhild 2013, p. 214). Thus, the position of the distant Other has changed, as Ndikung and Römhild (2013, p. 214) elaborated:

> Due to that spatial implosion the significant position of the distant Other has proliferated in a multitude of neighboring minorities vis-â-vis the respective majorities they constitute, including the diverse forms of "irregular" migrations emerging while crossing the new European borderlands, and the presence of postcolonial, post-migrant, post-socialist subjects and citizens as well as "dissident" genders, sexualities, subcultural, anti-neoliberal, post-capitalist political articulations and movements.

On the other hand, while this transformation increasingly shapes modern societies, it is being recognized in public discourse and politics at a slow pace. Confronted with the new and developing situation, the dominant politics of integration has reacted to this situation by insisting on traditional narratives and perceptions. They have had to increasingly overemphasize "constructions of an ethnicized, racialized Other in order to still keep up the fiction of a national, European, western domination over and distance from culturally inferior, marginalized subjects" (Ndikung and Römhild 2013, p. 214). Consequently, the political and cultural struggles about migration address the very idea of who Europe wants to be and which role the former distant Other will play in its present and future. According to Ndikung and Römhild (2013, p. 214), the figure of the post-Other emerges in this paradoxical moment of time, still bearing the "signs of historical Othering" while simultaneously "representing and experimenting with unknown futures beyond it".

Rasha Abbas' blending of various spaces and times as well as her surreal usage of dreamlike, hallucinatory spaces anticipated this moment of post-Otherness. The protagonists in her prose are not framed anymore as "distant Others", or as minoritized subjects who have to "integrate" or "arrive" in the new homeland. They are instead already part of

the new and emerging Europe. With the constant movement between different spaces and with the blending of different times, her stories disrupt the very logic of here and there as well as of integration and assimilation, which is historically built on the spatial dimension of colonial thinking. Further, Abbas' prose is anti-integrational in the sense that she does not accept the fundamental order of time and space, which the traditional European identity is built on. Rather, her characters unsettle the very distinction between us and them, "evanescing the border between the 'self' and the 'Other'", as Ndikung and Römhild (2013, p. 214) noted in relation to the most recent developments in contemporary arts.

Moreover, in their essay, Ndikung and Römhild also highlighted a few examples from contemporary art and culture, in which the aforementioned figure of post-Otherness appears and unfolds. The first example of the new artistic practices in contemporary art and culture, in which the new condition is expressed and reflected on, is the *postmigrant theater* in Berlin. In 2008, the theater director Shermin Langhoff along with cultural practitioners and activists took over Ballhaus Naunynstrasse, a small, independent theater in Berlin-Kreuzberg, which is a multiethnic and multireligious neighborhood. They labeled their work as *Postmigrantisches Theater* or Postmigrant Theater. In the following years, the theater became a major success among critics and audiences, offering a platform for new and other representations of migratory stories, which had not been staged in mainstream theaters in Germany (Sharifi 2011; Stewart 2017; Langhoff 2018; Petersen et al. 2019, pp. 33–35). With its artistic practices, the theater transgressed "the restricted space of 'ethnic minorities' towards 'native' mobile subjects" (Ndikung and Römhild 2013, p. 215). By doing so, they spoke "of and for an inclusive post-migrant Germany/Europe/world" (Ndikung and Römhild 2013, p. 215). The success of the concept continued when Shermin Langhoff in 2014 took over the long-standing and state-funded Maxim Gorki Theater in the center of the German capital, which has since emerged as the heart of postmigrant theater.

With this background in mind, it is therefore not surprising that the transformative power of literature and art was also at the center of the staging of Abbas' stories at the Maxim Gorki Theater in Berlin in the beginning of 2022. Specifically, the staging of her stories was part of the theater's attempt to present new stories of migration and disrupt traditional narratives. Instead of repeating the binary distinction between leaving and arriving and dislocation and relocation, the staging brought together various scenes from different stories, blending them into a multi-aesthetic and multi-language experience. The play, directed by Nübling (2022), consists of four actors, three of whom have a background in Syria themselves including Kenda Hmeidan. Additionally, it blends different forms of expressions, such as techno music and dancing scenes as well as monologues and playful performance of mimetic expressions. During the play, excerpts of Abbas' stories are quoted by actors, voiced through a telephone, or shown on text boards in Arabic, English, or German. Additionally, the four actors have different ways of dealing with the difficult and often surreal character of Abbas' prose: one actor, for example, uses changing and voiceless face-expressions to illustrate the feelings of the narrator; the dancer Lujain Mustafa—who was educated at the classic ballet school in Damascus and who is living in exile in Berlin as well—uses dance performances as the main form of aesthetic expression for the scenes and emotions in the stories. Far from establishing the traditional theater of representation or developing a coherent story, the staging instead evokes a "nightmarish, trancelike and imaginative formation," as a critic recalled (Büsing 2022). Meanwhile, another critic noted that the play evokes "animations from computer-games, video-projections of destroyed cities and tunnels" (Müller 2022). The spectator's view repeatedly penetrates into the scenery and backdrops of the stage as if following the perspective of a running film camera. However, in this ride with the camera everything repeats itself, creating a repetitive loop of voices and themes (Müller 2022).

Additionally, the actors introduce themselves with differing names and ages at various points, often triggering confusion among the audience. The name that the actors use repeatedly is from one of Abbas' stories: "Samt" (Abbas 2018, p. 7). Further, the ages, the professions, and the situations constantly change: Sometimes, they are 30 years old

and at other times, 20, 17 or 24. Overall, the idea of one linear story of migration and exile, typically framed through the binary distinction between leaving and arriving and between dislocation and relocation, is replaced by a more complex and more opaque story of migration and its aftermaths. Like the short stories, the staging blends different phases of life and brings together times and spaces. Thus, there is no coherent character, and the story is not centered with one predominant narrative. This rejection of a coherent story is mirrored in the recurring and yet ironic announcement of the actors that they will soon, in the next minutes, give "a summary of *A Summary of Everything that Happened*" (Nübling 2022). Against the widespread public expectation of being presented with a coherent story of migration and flight, this "summary of a summary" never occurs. Since this is only announced but never given, the expectations are playfully rejected, which emphasizes the impossibility of presenting a consistent linear storyline or providing a reasonable picture of the experiences of war, displacement, and exile. Moreover, the persistent fixation of the migrant or refugee as the Other—who is defined through their experience of flight and exile—is substituted with the recognition of the indispensable multiplicity of memories, identities, and experiences within the same person. Memories from the revolution in Syria, from war, and from the flight are part of the play, as is the case with various experiences and memories of the time spent in exile. Thus, instead of reaffirming a one-dimensional and simplified image of the refugee or the migrant, this multilayered staging of the stories connects pieces of memories, surreal family scenes, and experiences of being subject to xenophobia and racism in the Berlin night life. It highlights the impossibility of concentrating the manifold experiences and memories into one "clearly defined point" (Wahl 2022). As a result, the theater event turns out to be an impressive, "confusing, multiperspectival and multilingual theater-evening" (Büsing 2022), which does not support the traditional logic of representation of "refugees". The play rather depicts an "abyss, that seems to constantly reproduce itself" (Müller 2022).

### 5. The "Art of Survival" as an Aesthetic Strategy

The staging of Abbas' prose is surely not only impressive because of its aesthetic expression of the spatial implosion and what Ndikung and Römhild (2013, p. 215) called the "dissident reality of post-Other conviviality". It is also because of its depiction of the horrors of war, displacement, and flight.[5] However, most importantly, the play never falls into the pitfalls of patronizing and victimizing the "refugee." Instead, the staging constantly focuses on the transformative power of aesthetic freedom and imagination. In the beginning of the play, this imaginative power is emphasized and verbally expressed. Before taking the stage, the actors are placed among the audience, from where they repetitively whisper "imagine, imagine, imagine" into microphones (Nübling 2022). As this meta commentary appears before the actors are on stage, the spectators are compelled to see the scenes of the play as possibilities and potentialities rather than as truthful and authentic depictions of experience of flight and exile. The "cult of authenticity," often applied to artists dealing with experiences of migration (Haakh 2013, p. 38), is thus explicitly rejected from the very beginning. The staging clearly widens the range of possible stories that refugees and migrants are expected to tell as does Abbas' surreal and partly hallucinogenic stories. In addition, the audiences are forced to "read" the play in a different and more complex way. Based on French philosopher Rancière's (2009) writing, one can even say that the spectators are forced to find their own way through the impressions, images, and associative pictures presented during the play. Instead of being presented a singular story or message and meaning, the spectators are made to explore individualized paths through the multiplicity of images and impressions (Rancière 2009, pp. 1–23).

Moreover, the notion of "survival" is at the center of the play constantly. Specifically, the actors repeatedly lean forward against the imaginative power of a strong wind coming against them while chanting, "survival, survival, survival." In those scenes, the combination of techno-music, dancing scenes, the experiences of insecurity and vulnerability in the asylum system as well as memories of war and flight merge into the subject's longing to

stand against the power of history and of external identity ascriptions such as those by migration and border regimes. In this context, the repeatedly performed scene illustrates the longing to stand against the influence of power regimes, ascribed identities, and various forms of oppression.

Abbas' emphasis on the aesthetic transformation of the reality as well as the recurring references to the notion of "survival" can be understood through the concept of *Überlebenskunst* or the "art of survival." This concept has been highlighted in more recent migration studies in relation to the question of agency (Seukwa 2006; Hill 2019; Schacht 2021). It is based on the development of autonomous strategies by refugees and forced migrants who are constantly confronted by the power of migration regimes. The concept expresses the idea that migrants and refugees are not only objects of those migration regimes and power oppression but also actively develop strategies of agency and autonomy. It is particular interesting, as Schacht (2021, p. 20) notes, "how humans under restrictive societal circumstances develop creative strategies und life-concepts, in order to position themselves in the society". According to this perspective, while refugees and migrants are influenced by the power relations and social limitations that they are confronted with, they are also actively reacting against those restrictions and limitations. Thus, it is important "not (only) to conceive persons with migratory and refugee experiences as objects of external representations and powerful discourses, but to understand the creation of their own strategies as active reaction against the circumstances at the borders" (Schacht 2021, p. 21). The term *Überlebenskunst*, or art of survival, mirrors these strategies to regain agency in a difficult power relationship. Migration researcher Seukwa (2006), who introduced the notion of *Überlebenskunst* in academia, went on to define the "habits of the art of survival" that refugees develop during forced migration.

Thus, refugees and migrants do not submit themselves as innocuous victims to border regimes. Instead, they employ "multiple tactics and tricks," both in relation to the challenges of flight and the struggles in the new homeland's asylum system, which is typically "dominated by alienation, social isolation and incessant insecurity" (Seukwa 2015). When read from this perspective, the stories of Rasha Abbas illustrate that the aforementioned strategies of survival are strongly related to the concept of aesthetic freedom and the transformative power of art. It can even be said that the very concept of "art of survival" appears to be an aesthetic concept, as it presumes the aesthetic imagination of new realities and possibilities, potentially expanding the given version of reality. The post-Other as a struggle to survive and regain control beyond traditional identity ascriptions needs the imaginative power of art and culture in order to unfold visions of "unknown futures" beyond the historical state of Otherness (Ndikung and Römhild 2013, p. 224). Accordingly, the artistic practice of "juxtaposing the apparently unknown with the apparently known" also poses the question of "how to address and to make space for 'difference'" in a society that is still dominated by homogenization discourses (Ndikung and Römhild 2013, p. 224).

By rejecting traditional logics of representation and working towards the conflation of various spaces and times in artistic images, Abbas reflects on the complexities of contemporary subject positions. While the harshness of war, exile, and exclusion is clearly expressed in her prose, these experiences are no longer presented as markers for the "distant Other". Instead, Abbas' aesthetics of the "art of survival" can be viewed as an outspoken and complex artistic articulation of "post-Otherness": a mode of existence where the experiences of humiliation, war, and flight are part of the collective memory of an emerging "postmigratory normality" (Ratković 2018, p. 129).[6]

## 6. Postmigratory Normality

In her essay from 2016, Dahinden mentioned three strategies of how to deal with the methodological challenges of contemporary migration studies. First, the acceptance that the migration research originates in what she calls a "migration apparatus" that works within the field and the system of power relations connected to it (Dahinden 2016, p. 2212). Second, a "strategic positive essentialism" could be employed using the commonly

accepted distinction between migrants and non-migrants in order to address and challenge certain forms of inequality (Dahinden 2016, p. 2212).[7] Third, a reflexive attitude could be pursued by searching for new ways to approach migration research. This would imply, as Dahinden (2016, p. 2212) explained, an investigation into "methodological strategies that make it possible to de-naturalize and to de-ethnicize migration and integration studies". Her own approaches contribute to this reflexive attitude, by, among others, opening the field of migration studies to the use of general sociological theories as well as examining the contexts of traditional distinctions between the migrants and non-migrants—that is, "exploring when, how, and on behalf of which markers that specific boundaries between us and them are established, transgressed or dissolved, and what consequences such boundary processes may have" (Dahinden 2016, p. 2216). As per this perspective, migration scholars would no longer be specialists in migration and integration issues but rather social and cultural scientists who focus on processes of migration and alienation as part of their research in social and culture negotiations (Dahinden 2016, p. 2218). Clearly, this third position is in line with Römhild's (2017, p. 70) statement, quoted above, that we do not need more research in migration but rather migration-based insights into the "contested areas of 'culture' and 'society'".

When considered against this background, Rasha Abbas' stories can be seen as depicting the harsh reality of war and exile but also allowing for new and broader perspectives and complexities. Instead of reading her stories as mere depictions of the experiences of war, flight, and exile—which would potentially cement her to the heritages of the past—and instead of reducing her work to expressions of a supposedly authentic and victimized Other, they should be seen as reframing the very experiences of migration and flight and integrating them into a new, complex picture of contemporary societies. With her combining of time and spaces, she allows us to perceive the aforementioned experiences as part of a conflictual but emerging post-Other-Europe: where different backgrounds and experiences stand side by side, and the multitude of minorities interact instead of being positioned at the margins, outside the imaginary center of society. By doing so, Abbas' stories force us to reconsider the logic of "migrantology" and to explore new and different paths in the scholarly discussions on the representations of displacement, migration, and flight. While the traditional representation of refugees as either victims or threats tend to place them outside of the normality of postmigrant societies, Abbas' surreal and hallucinatory stories instead move the experiences of war and exile into the center of society. In doing so, the stories support a novel understanding of contemporary societies, which has been discussed through the concept of "radical diversity" (Czollek et al. 2017).

According to this concept, the diversity of "backgrounds" is not reserved for persons with backgrounds in migration but is rather an inherent part of modern societies' multiplicity. In a short and programmatic text, published in a monthly program flyer by the Maxim Gorki Theater (2016) in Berlin, this expanding of the notions of backgrounds is put forward. Specifically, the flyer confirms that there has been a lot of talk about "backgrounds" when speaking of the Gorki Theater, which is an unmistakable reference to the widespread use of the notion of "immigration backgrounds" in the public discourse in Germany.[8] In the next line, however, the flyer expands the perspective by saying that it is true that the people working at the Gorki Theater have "biographies that go far beyond their own lives," but the flyer notes that this simply affirms that they are "just like everyone else" (Gorki Theater 2016). Thus, instead of reserving the notion of "backgrounds" for only one part of the population, as it is done in large parts of the public debate that revolves around the notion of "immigration backgrounds," the concept is applied to everybody. Consequently, the flyer then mentions the various backgrounds of people working at the theater, including family backgrounds inside and outside Germany as well as different social, sexual, religious, and political backgrounds (Gorki Theater 2016).[9] Finally, the flyer ironically states that this is just as unbelievable as it is real before moving on to note that the background of a person is not necessarily the decisive factor for artistic practice at the Maxim Gorki Theater. Instead, the background of a person only "becomes a statement when they themselves decide if,

when, and most importantly, how it should be told" (Gorki Theater 2016). Against the persistent tendency of attaching "migrants" and "refugees" to their "background", the Gorki Theater focuses on a person's autonomy of how and when to address their individual backgrounds and experiences of the past.

Here, it is important to note that Rasha Abbas' stories and her aesthetic strategy of the "art of survival" feeds into this way of thinking. Her stories not only challenge traditional, hegemonic narratives about migration and flight and substitutes representations of victimhood with the agency of aesthetic freedom, but also envisions a realm of radical diversity where different backgrounds and times and spaces stand side by side and interact with each other. Fighting the persistent power of binary distinctions and separating "migrants" and "refugees" from a supposedly non-migratory we-group, her approach is insistent on the post-Otherness of modern existences. Through the "art of survival" as an aesthetic strategy, the past becomes an undeniable part of the present and future and is thus part of a developing postmigratory normality. Against the widespread cult of authenticity, which expects the "refugee" and the "migrant" to tell a coherent, personalized, and realistic story, it is the non-authenticity, the non-personal, and the non-realistic approach that allows Abbas to create spaces of aesthetic negotiation and participation where the experiences of the past can become a part of present and the future identities. At the core of this aesthetic strategy surfaces a utopian vision of post-Otherness, as traditional approaches and perspectives are replaced with "formerly marginalized forms of knowledge practices" (Yildiz 2022, p. 47). Moreover, the transition from being a refugee to becoming a citizen, which Hmeidan (2022, p. 4) identified as common task for all Syrian refugees living in exile, is not taking place through the logic of integration or assimilation, but rather through the conflation of the distances and times between "us" and "them" and between the Other and the imagined community of non-migratory Europe. Thus, the formerly distant Other has moved into the center of society. Accordingly, the idea of "dancing with the sniper" is not the dream of the Other but instead an inherent part of our common present and future.

**Funding:** This research received no external funding.

**Institutional Review Board Statement:** Not applicable.

**Informed Consent Statement:** Not applicable.

**Data Availability Statement:** Not applicable.

**Conflicts of Interest:** The author declares no conflict of interest.

## Notes

[1]   The "reflexive turn" is at least partly a consequence of the debates in cultural studies in England and the US in the late 1980s. In particular, the attempts to de-essentialize concepts such as "culture" and "ethnicity" influenced the field, emphasizing the necessity to reframe some of the traditional approaches and perspectives (Hall 1997). Some of the most recent developments in migration studies, such as the emerging concept of "postmigration," are in reaction to those fundamental changes by creating space for new reorientations (see Baumann and Sunier 1995, regarding the historical development, see Gaonkar et al. 2021, pp. 14–16).

[2]   Dahinden's essay from 2016 was a reaction to the original version from Römhild, which was published in German in 2015 (Römhild 2015). The English version came out in 2017 in themed issues on "(Post-)Migration in the Age of Globalisation" (Petersen and Schramm 2017). The plea for the demigrantization of migration research goes back to the discussions in "Project Migration" from the Humboldt University in Berlin (see Römhild 2017, p. 70).

[3]   In an interview with Clara Hermann, Rasha Abbas refers to the dreamlike nature of her stories, saying that she usually follows "a dream's logic to build a story." She goes on to then address the intercultural and transnational dimensions of dreams and the "collective subconscious," which she believes exists despite all cultural differences and individualized historical backgrounds. Accordingly, employing a dreamlike logic both entails traces of a culturally-determined field of references—as dreams are also working with images and pictures inspired by various cultural histories and traditions—and deconstructs them by focusing on the "collective subconscious," which transgresses cultural spaces and times (Abbas 2015).

[4]   In her book, Rebentisch (2016) critically engages with the widespread stance against the aestheticization of politics and aesthetic performativity, as put forward by numerous philosophers and political thinkers since Plato. She then goes into detail and shows that this critique of aestheticization is historically conceptualized around the fear that the theatralization of politics and personal

identities would have a "disintegrating effect" not only on politics but also on the "political community" (Rebentisch 2016, p. 7). According to the position of the critics of aestheticization, the supposedly close and naturally born social bonds between members of society would be replaced by nothing more than "'aesthetic' relations," destabilizing the political community (Rebentisch 2016, p. 7). However, in her own examination of the relation between aesthetics and the political, she comes to a different conclusion. The distance from the social, as it appears among others in the self-transformations of the so called "Lebenskünstler" with their aesthetic lifestyle, does not necessarily entail "a kind of distance from all social determinacy that is as abstract as it is imaginary" (Rebentisch 2016, p. 9). Rather, it is the overall mutability of the social, which is expressed in aesthetic existences and dazzling life-forms. When seeing it this way, the aestheticization of freedom can no longer be considered the "misunderstanding of a kind of freedom from the social in a kind of non-dialectic opposition to freedom in the social." Rather, it expresses "the tension at the heart of every individual's life" (Rebentisch 2016, p. 9). Accordingly, every change in social norms and values takes its point of departure in the individual's experience of self-difference, which "compels the subject to reconceive of itself, its self-understanding, and the meaning of its subjectivity from a distance" (Rebentisch 2016, p. 9). The democratic existence, she notes, is founded in this experience of "aesthetic freedom," as it highlights and makes possible the changeability and mutability of the given social reality (See also Rebentisch 2007, 2013; Schramm 2015).

[5] In a similar way, Rebentisch pointed out the importance of content for the aesthetic experience. According to her, aesthetic experience is dependent on both the aesthetic form of the images we are presented with, and the content that those images engage. Aesthetic experience, "seems only to be able to gain a certain intensity and thereby quality if the contents, that are brought into the aesthetic play, matter for the experiencing subject" (Rebentisch 2007, p. 63).

[6] The concept of "postmigration", including the notion of the "postmigrant society," has gained traction in academic circles during the last approximately 15 years. The concept does not denote and signalize the end of migration and thus not refer to an already existing state where the exclusion of "migrants" and "refugees" is overcome but instead to a conflictual space of struggles and negotiations, occurring after migration has taken place (on the different conceptualizations of the term, including the criticism against it, see Petersen et al. 2019, pp. 11–24, 50–63; Gaonkar et al. 2021, pp. 17–25; Foroutan 2019a, 2019b).

[7] This approach is mirrored in traditional concepts of "migration literature." As mentioned above, this concept emerged historically as an attempt to increase the visibility of formerly marginalized voices. Till today, the approach can be used as a form of "strategic essentialism" (Gayatri Chakravorty Spivak) to overcome patterns of exclusion and marginalization. At the same time, the following question arises: does this approach reaffirm binary distinctions and symbolic demarcation lines between the works of the "migrant" and non-migratory we-group? (see the discussion in the beginning of this chapter).

[8] The concept of "migration background" (*Migrationshintergrund* in German) was introduced in Germany in the beginning of the 21st century for the statistical purpose of "counting" the percentage of immigrants and their descendants in German society. Today, the concept is controversial in discussions. However, it is broadly used in public discourse, as apparently everybody knows and uses it (on the different definitions of "descendants" from an international perspective, see Supik 2014).

[9] See on the following reading of the flyer also Moslund et al. (2019, pp. 242–43).

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
