# Peer review of "Dancing with the Sniper: Rasha Abbas and the “Art of Survival” as an Aesthetic Strategy"

_humanities, doi:10.3390/h12020029_

Round 1

Reviewer 1 Report

The essay is strongly argued, theoretically sophisticated, engages compellingly with a complex author and with a significant body of migrant studies discourse.

There is one overriding concern that cuts through my evaluation of the piece:

How immersed is this author in the Arabic originals of Rasha Abbas's fiction? I do not mean this as a purist's philological reproach: Abbas's fiction certainly merits consideration as world literature circulating in translation. However, if the author only reads her in translation - he/she needs to make this clear and incorporate some critical reflection on the meaning of reading her work in translation. If the author can read and has read the Arabic, a more deliberate reflection is in order on the changes undergone by the texts in their German translations and adaptations for the stage.

Assuming Arabic knowledge, a way should be found to make the texts themselves speak and make Abbas's style more palpable in the essay. Less labels such as "surrealistic" or "hyper-realist" and more actual texts would be preferable. A lot is at stake, since the more absent Abbas's voice is from the essay, the more suspicion is raised that her texts are being co-opted into an internal debate in migration studies and thus feeding "Western" theory.

The lack of framing with respect to Syrian and Arabic literature is understandable if the author comes from a non-Arabic discipline. It is slightly risky to make her appear as the representative Syrian author without any literary history and collective literary frame of reference. A clear instance where intertextual awareness would enhance the reading is the reference to the sniper - which to me appears a reworking of a well-known passage in the novel "Story of Zahra" - whose plot takes place during the Lebanese civil war - by the Lebanese author Hanan al-Sheikh. Suggest looking up the reference and incorporating some thoughts on it. 

On another note, I am not sure that Abbas's fiction can serve as an adequate representation for the experiences of the "boat people" and how they perceive themselves. Highly recommend consulting the work of Syrian poet Luqman Derki in this respect.  It is very tough to find a representative voice for the displaced Syrians since the experience is by nature extremely fragmented and is all about fragmentation. So I advise taking some caution and adding the necessary reservations about making her a stand-in for the whole. 

Otherwise, impressive and engaging essay. Well done!   

Author Response

Dear reviewer, 

I would like to thank you very much for the detailed and constructive review of my manuscript on the works of Rasha Abbas. Your comments were very much appreciated, and very helpful! As I am - unfortunately - not able to read Arabic, I had to read Abbas' works in German and English translation, which obviously is a challenge. I now did address the issue in the revised version, as you correctly point to the fact that this should be mentioned and discussed in one or the other way. Your comment made it even more clear to me, that I actually read Abbas' works more as part of the ongoing negotiations on belonging etc. within Europe, than reading her as part of the Syrian and Arabic literature tradition (which would be extremely interesting as well, but unfortunately outside of my knowledge). In this context, your comment to Hanan al-Shaykh was, however, particular helpful. I looked it up and not at least the scene of "dancing with the sniper" seems indeed very much to rework some elements of the Story of Zahra, which I have not been aware of. I added some lines on it, which hopefully improve my reading of this specific "dancing"-image and on Abbas' novel as such. It could even be interesting to take a closer look on this interextual relation in another context. For now, once again: Thank you very much for taking your time and reading my essay, it was really helpful indeed. 

All the best

the author

Reviewer 2 Report

The essay is excellent; well-researched, relevant, and organized. My only question/concern is that there is no mention and/or consideration of the Turkish community in Germany, especially in Berlin and the neighborhood of Kruetzberg where the theater is. From my limited experience of this neighborhood and the history of Turkish guest-workers in Germany, their long presence in the city, but that neighborhood specifically would play into the community's and scholars' attitudes and new-thinking in regards to migrants and self-autonomy. I understand the essay specifically discusses Abbas' work, but I think there deserves to be recognition and acknowledgement of the Turkish (and perhaps others) community beyond a passing statement on the diversity of the neighborhood.

Author Response

Dear reviewer

Thank you very much for the positive and constructive review, which was very helpful indeed! I considered your comment about the local context of the theatre work, which is very relevant, but decided in the end not to include more reflections on the history of the area of town. While it is true that the theatre work at the Ballhaus Naunynstrasse in Berlin (which I mention a couple of times) grew out of the local community in Berlin Kreuzberg, including many people of Turkish descendant, the play itself, which I am discussing in the paper, is staged at another theater, the Maxim Gorki Theater, which is not located in this specific area of town, but rather in Berlin-Mitte, very close to Unter den Linden and the center of the town. I therefore decided not to engage in further discussions on the location, as it would mainly be relevant for the Ballhaus, and not the Gorki Theater, which has my main interest in the essay.